# Multiplexed Quantitation of Plasma Proteins by Targeted Mass Spectrometry for Early Diagnosis of Pancreatic Ductal Adenocarcinoma

**DOI:** 10.3390/ijms26189219

**Published:** 2025-09-21

**Authors:** Dmitry N. Korobkov, Ivan A. Molodtsov, Alexey S. Kononikhin, Anna E. Bugrova, Maria I. Indeykina, Alexander G. Brzhozovskiy, Dmitry Yu. Kanner, Evgeny N. Nikolaev, Elena Vasilieva, Alexey A. Komissarov

**Affiliations:** 1Moscow City Oncology Hospital No. 62, 143515 Krasnogorsk, Russia; 2I.V. Davydovsky Moscow City Clinical Hospital, Moscow Department of Healthcare, 117463 Moscow, Russia; 3Project Center of Omics Technologies and Advanced Mass Spectrometry, 121205 Moscow, Russia; 4Emanuel Institute for Biochemical Physics, Russian Academy of Science, 119991 Moscow, Russia; 5Laboratory of Atherothrombosis, The Russian University of Medicine, 127473 Moscow, Russia

**Keywords:** pancreatic cancer, pancreatic ductal adenocarcinoma (PDAC), diagnosis, biomarkers, proteomics, mass spectrometry

## Abstract

Pancreatic cancer (PC) is the sixth leading cause of cancer-related deaths worldwide. Patients with pancreatic ductal adenocarcinoma (PDAC), the most common type of PC, have a 5-year survival rate of approximately 10%. This low survival rate is mainly attributed to late-stage diagnoses and the lack of robust screening methods. Several serum proteins have been proposed as potential PDAC biomarkers, but they have not been introduced into clinical practice due to their low sensitivity and specificity. Therefore, the identification of new PDAC biomarkers remains highly important, and multiple reaction monitoring (MRM), a highly accurate mass spectrometry (MS) technique, can be used for this purpose. Using MRM MS analysis, we estimated the concentrations of 103 proteins in peripheral blood plasma from 132 participants: patients with newly diagnosed PDAC at different stages and healthy individuals. We identified six proteins that were differentially presented between healthy controls and patients with PDAC at all stages (adjusted *p*-value < 0.01), and that were associated with survival rates for 23 months. A developed cross-validated model based on these six proteins showed an average accuracy of 90% in distinguishing between early-stage PDAC and healthy controls (AUC = 0.933). However, further research is needed to implement this model in clinical practice.

## 1. Introduction

According to world-wide statistics, pancreatic cancer (PC) is one of the most aggressive cancers, as it is associated with a high annual number of deaths [1,2,3]. The most common type of PC is pancreatic ductal adenocarcinoma (PDAC), comprising approximately 90% of all PC cases [4]. The 5-year survival rate for PDAC is approximately 10%, which is mainly because of late diagnosis—most patients present with either unresectable locally advanced or metastatic disease, when currently available cytotoxic therapies are modestly effective [1,2,3,4]. In this regard, the most successful approach to improving survival rates appears to be the early detection of PDAC. However, this is a difficult task, as early-stage curable disease is often asymptomatic. Therefore, there is a pressing need for specific biomarkers for this disease.

To date, several serum proteins have been suggested as PDAC biomarkers, including serum carbohydrate antigen 19–9 (CA 19-9) [5], keratin 8 [6], protein induced by vitamin K absence II [7], and gremlin 1 [8]. Among them, only CA 19-9 has been used in clinical practice for diagnosis and monitoring of treatment response. However, due to low sensitivity and specificity, its value as a screening tool is being revisited (discussed in [9]). For this reason, multiplexed screening of blood proteins is highly required for the search and validation of potential PDAC biomarkers.

Multiple reaction monitoring (MRM) is a highly precise mass spectrometry (MS) technique used in proteomics to perform comprehensive quantitative analyses and validation of potential disease biomarkers [10,11,12,13]. Recently, the MRM assay was developed and validated for multiplexed screening of up to 241 human plasma proteins [14,15], including 61 FDA-approved protein biomarkers [16] as well as potential biomarkers of PDAC [17,18]. The assay demonstrated high accuracy and reproducibility for selected plasma protein quantification, which is essential for further clinical proteomic research.

Here, using MRM MS analysis, we estimated the concentrations of 103 proteins in the peripheral blood plasma from patients with newly diagnosed PDAC at different stages and healthy individuals. We identified six proteins that are differentially presented between healthy controls and patients with stage I PDAC, and which are also associated with survival rates at 23 months post diagnosis. These proteins could potentially serve as early diagnostic and prognostic biomarkers for PDAC.

## 2. Results

### 2.1. Patients’ Characteristics and Follow-Up Survival

In total, 132 participants have been enrolled in the study. Among them, there were 113 patients with newly diagnosed PDAC at different stages and 19 healthy controls. The tumor was localized mainly in the head (n = 76, 67.3%) and body (n = 29, 25.7%), and rarely in the tail (n = 8, 7.1%) of the pancreas. The majority of clinical parameters, including common confounders like age, sex, body mass index, etc., showed no statistically significant difference between the groups, while those that differed fell within the normal range (Table 1).

After the enrollment, patients with PDAC were followed up with for 23 months to monitor all-cause mortality. Among 101 (89.4%) patients with known outcomes, we registered 58 (57.4%) lethal cases (Table 1). The Cox model was used to estimate the hazard ratio (HR) of mortality in relation to clinical stage. Stage IV was found to be significantly associated with a high risk of mortality (Figure 1A). These results were validated through a Kaplan–Meier analysis of overall survival: patients with stages II-IV had lower survival rates, while those with stage I had the highest ones (Figure 1B). These findings align with the reported survival rates for PDAC patients and indicate that early detection is a promising approach to improving disease outcomes.

### 2.2. Potential Protein Biomarkers of PDAC Among Plasma Proteins

Multiplexed quantitative proteomic analysis of 103 plasma proteins was performed for all participants using the MRM MS approach. A comparison of the protein profiles between healthy controls and stage I PDAC patients identified 30 differentially expressed proteins as potential early biomarkers of the disease (Appendix A). Additionally, a receiver operating characteristic (ROC) curve analysis was performed to estimate the applicability of these proteins for classifying healthy individuals and stage I PDAC patients. The revealed precision and recall parameters characterized all identified proteins as relatively robust classifiers, while 24 proteins were distinguished by an area under the curve (AUC) higher than 0.8. Notably, three apolipoproteins (A-IV, F, and A-II) were identified among the proteins with AUC > 0.8. However, PDAC patients are known to suffer from significant weight loss, especially at late stages [19]. For this reason, as apolipoproteins are directly influenced by diet, they were excluded from further analysis. In total, 21 proteins have been identified in peripheral blood plasma as potential early biomarkers of PDAC (Table 2).

Next, the HRs of mortality in relation to concentrations of these 21 proteins were estimated within the PDAC group using the Cox model. Six proteins were found to be significantly associated with the survival rates: higher concentrations of alpha-1-acid glycoprotein 1, beta-2-microglobulin, plasma protease C1 inhibitor, and leucine-rich alpha-2-glycoprotein were associated with the increased risk of lethal outcomes, while high fibronectin and cholinesterase concentrations, in contrast, lowered the risk (Figure 2A). Furthermore, concentrations of these proteins were found to be increased or decreased, respectively, in the peripheral blood plasma of PDAC patients at all stages compared to healthy individuals (Figure 2B). This makes these proteins the «universal» PDAC biomarkers for both early and late-stage diagnosis.

A Kaplan–Meier analysis confirmed these results. All PDAC patients were divided into two equally represented groups by the median value of plasma concentration of the identified proteins. The survival rates significantly differed between the groups in accordance with the calculated HRs for each protein (Figure 2C).

### 2.3. A Panel for Early-Stage PDAC Diagnosis Based on the Plasma Protein Biomarkers

A model that takes into account the levels of these six proteins in blood plasma to distinguish between stage I PDAC patients and healthy donors was developed. The proposed cross-validated model discriminates early-stage PDAC patients with an accuracy of 0.90 ± 0.09, precision of 0.82 ± 0.17, recall of 1.00 ± 0.00, specificity of 0.85 ± 0.14, and an AUC of 0.933 ± 0.063 (Figure 3).

Taken together, the results of the current study demonstrate that alpha-1-acid glycoprotein 1, beta-2-microglobulin, plasma protease C1 inhibitor, leucine-rich alpha-2-glycoprotein, fibronectin, and cholinesterase are potential biomarkers for early diagnosis of PDAC.

## 3. Discussion

According to the project GLOBOCAN 2018 data, pancreatic cancer (PC) was ranked the 11th most common cancer, causing 4.5% of all cancer deaths in 2018 [20]. Since then, the incidence and mortality of PC have increased [1,2,3]. Currently, PC is classified into four stages: at stage I a tumor has no spread and, thus, is resectable; at stage II a tumor has spread locally to the nearby lymph nodes but is still borderline resectable; at stage III a tumor commonly expands to the nearby blood vessels or nerves, but has not metastasized to distant sites, however, it is becoming unresectable; at stage IV a tumor spreads to distant organs and is becoming metastatic. The most common type of PC is pancreatic ductal adenocarcinoma (PDAC), comprising approximately 90% of all PC cases [4]. At early stages, PDAC usually lacks symptoms, but upon progression, the disease is characterized by the onset of nonspecific symptoms, including jaundice, weight loss, abdominal pain, fatigue, and cholangitis [21,22]. As a result, PDAC is typically diagnosed at a late stage (III or IV), and, therefore, has a poor prognosis. Surgery, chemotherapy, radiotherapy, and their combination are options used to increase the survival rates of PDAC patients; however, advanced-stage PDAC has no definite and efficient therapy. New treatment approaches are currently being tested, including novel neoadjuvant therapies, CAR-T immunotherapy, and anti-cancer vaccines (discussed in [23]). Unfortunately, these approaches possess modest overall benefits and have shown variable efficacy. In this respect, the diagnosis of PDAC at early stages is critical for curative treatments and may be the key to improving the disease prognosis.

The screening of the general population for early detection of PDAC is currently not feasible, primarily due to the lack of reliable and robust biomarkers. However, the studies aimed at the screening of targeted groups, specifically with family history, demonstrated that several molecules in blood and pancreatic juice have the potential to serve as markers for PDAC (discussed in [24,25,26]). Blood-based markers are the most effective for broad screening. As the concept of “liquid biopsy” suggests, blood is a body fluid that can be obtained with minimal invasiveness and yet provides a wealth of information about the organism [27,28]. Among blood-based molecules, the serum carbohydrate antigen 19-9 (CA 19-9) is the sole biomarker that has been proven to be clinically beneficial. It has been employed in clinical settings for the purpose of treatment monitoring and detecting relapses at an early stage. However, the use of CA 19-9 as a screening tool has had disappointing results, primarily because of false positives in conditions of inflammation and nonpancreatic cancers and false negatives in Lewis-negative individuals [29]. Other potential blood-based biomarkers of PDAC include proteins, circulating antibodies, PC specific metabolites, and miRNAs (discussed in detail in [26]). They exhibit a degree of specificity and sensitivity that varies between 45 and 95%. However, in these approaches, high specificity is commonly associated with low sensitivity, and vice versa. In this respect, the most effective approach was found to be the serum proteins’ signature, which comprises 29 biomarkers and enables the detection of stage I-II pancreatic cancer in humans with an AUC of 0.96 and sensitivity/specificity of 95/94%, respectively [30]. Despite the encouraging outcomes, the panel of 29 proteins appears to be excessively complex, and further testing on different populations with diverse ethnicities and genetic profiles is necessary.

The objective of the current research was to discover novel potential early-stage markers of PDAC among the proteins presented in the peripheral blood plasma. To achieve this, we employed a multiple reaction monitoring (MRM) mass spectrometry approach. This technique offers several advantages, including high accuracy and specificity in quantitative analysis. Additionally, ready-to-use commercial kits for MRM MS enable the processing of thousands of samples per week, with the use of appropriate instrumentation and automation, including faster liquid chromatography gradients and accelerated protein digestion within 5–10 min, implementing automated liquid handlers and simplified protocols to reduce hands-on time. All these features make MRM MS an effective tool for validating potential PDAC biomarkers and translating them into clinical diagnostics. Here, we used MRM MS to compare the protein profiles of healthy donors and PDAC patients at stages I-IV. We identified proteins that are differentially presented in blood already at stage I and whose levels are associated with survival rates. Moreover, these proteins exhibit a high degree of precision and accuracy in distinguishing between healthy individuals and those with early-stage PDAC. The proteins are alpha-1-acid glycoprotein 1, beta-2-microglobulin, plasma protease C1 inhibitor, leucine-rich alpha-2-glycoprotein, fibronectin, and cholinesterase.

Among these proteins, alpha-1-acid glycoprotein 1 has already been proposed as a potential biomarker for PDAC [31]. The same observation was independently obtained in our research, but we have also demonstrated the potential of this protein for early-stage detection. Another protein, plasma protease C1 inhibitor, was suggested as a PDAC biomarker within the panel of 29 proteins discussed above [30]. Additionally, plasma protease C1 inhibitor has also been identified as a potential biomarker for early diagnosis and prediction of Alzheimer’s and Parkinson’s diseases [32,33]. Similarly to plasma protease C1 inhibitor, leucine-rich alpha-2-glycoprotein, in combination with CA19-9 and tissue inhibitor of metalloproteinase 1, exhibited a high performance in the detection of early-stage PDAC [34,35]. However, leucine-rich alpha-2-glycoprotein is known to be involved in a wide spectrum of pathological conditions, including different types of cancer (discussed in [36]), which raises the question concerning the specificity of this protein as a marker for PDAC diagnosis. It is the same for beta-2-microglobulin. We did not find reported associations between beta-2-microglobulin levels and PDAC, although it was shown in vitro that the protein may be involved in the regulation of pancreatic cancer cells’ migration [37]. However, a wide range of studies have demonstrated that the levels of beta-2-microglobulin in the blood are significantly elevated in patients with renal disease, hematological malignancies, several solid tumors, and autoimmune diseases (discussed in [38]). Due to the involvement of leucine-rich alpha-2-glycoprotein and beta-2-microglobulin in various types of cancer and other pathologies, it is not advisable to use these proteins individually for the specific diagnosis of PDAC. However, when combined with more specific protein biomarkers, they can enhance the accuracy of PDAC detection.

In contrast to proteins discussed above, fibronectin and cholinesterase were found to be inversely correlated with the survival rates in patients with PDAC in our study. In agreement with this finding, low levels of cholinesterase in serum have already been reported to be associated with the poor prognosis of pancreatic cancer [39] and other tumors [40,41,42]. However, currently available information concerning the role of fibronectin in tumor genesis and progression is highly controversial. Particularly, high levels of fibronectin in blood or urine could be detected in late-stage metastatic cancers, while fibronectin suppression accompanies or even mediates tumorigenesis (discussed in [43]).

Taken together, the proteins we have identified in our study as potential biomarkers for early-stage PDAC have been previously associated with PDAC and other tumors. However, our study is the first to combine these proteins in a single panel and investigate them simultaneously. As a result, the developed panel demonstrates a high performance for distinguishing early-stage PDAC from healthy controls, with an average accuracy of 90% and an average AUC = 0.933. As discussed above, currently the highest performance among blood protein-based approaches for the detection of early-stage PDAC in humans has been demonstrated for the 29-protein panel [30]. The protein panel proposed in our study consists of six proteins, with only one protein being shared with the 29-protein panel—plasma protease C1 inhibitor. In this respect, our panel is more compact while having comparable quality parameters.

Nevertheless, the panel needs further validation due to the limitations of the current study. The study was conducted with a relatively small number of participants, so the applicability of individual proteins and their combined panel for early-stage PDAC diagnosis should be confirmed using an expanded cohort of PDAC patients and healthy controls. However, within the study, we controlled the false discovery rate, and the significance levels were found to be rather high (adjusted *p*-values ~10^−3^). All this suggests that the results obtained are pronounced enough to be detected even on a relatively small number of samples, and, therefore, they are likely to remain reliable when the size of the analyzed cohort is expanded. Additionally, since the proteins identified in the study as PDAC biomarkers have already been associated with tumors and other pathologies, to ensure the specificity of the diagnosis, the panel should be analyzed using other cancer types and non-cancerous conditions. Due to the reasons discussed above, our subsequent study will be a randomized multicenter prospective research aimed at the validation of the developed six-protein panel. We are planning to include patients with a wide range of pathologies, other than PDAC, to check the specificity of the individual panel and, in combination with other biomarkers, particularly CA19-9, as well as to prospectively track the incidence of PDAC among a cohort enrolled to validate the prognostic value of this panel.

## 4. Materials and Methods

### 4.1. Participants’ Enrollment and Peripheral Blood Plasma Isolation

Patients with newly diagnosed and histologically confirmed PDAC were enrolled in the study in Moscow City Cancer Hospital No. 62. The exclusion criteria were disease origin due to metastases in the pancreas from another tumor, and the patient’s refusal to sign the informed consent. Healthy individuals (those without active inflammatory processes, any infectious diseases, and any oncological diseases that have been diagnosed at the time of enrollment or in the anamnesis) were enrolled in I.V. Davydovsky Moscow City Clinical Hospital among individuals who underwent a routine health check-up. All participants underwent physical examination, as well as standard laboratory tests, including complete blood count and biochemical blood test, using automatic analyzers according to the manufacturers’ standard protocols.

Venous blood samples were obtained upon hospitalization. Peripheral blood was collected into S-Monovette 2.7 mL K3E tubes (Sarstedt, Nümbrecht, Germany) through venipuncture. Plasma was obtained by centrifugation of the tubes at 4000× *g* for 10 min and stored at −80 °C.

### 4.2. Sample Preparation for Proteomic Analysis

Plasma samples (10 μL) were processed for proteomic analysis according to a standard protocol [44]. Briefly, proteins were denatured and reduced in a solution containing 9 M urea, 20 mM dithiothreitol, and 200 mM Tris-HCl at pH 8.0 for 30 min at 37 °C, followed by alkylation with 100 mM iodoacetamide in the dark for 30 min. Trypsin digestion was then performed by first diluting the sample to reduce the urea concentration below 1 M, adding TPCK-trypsin at a 20:1 protein-to-enzyme ratio, and incubating at 37 °C for 18 h. The digestion was stopped by acidifying the samples with formic acid to a final concentration of 1.0%.

A quantitative assay was developed to target 103 plasma proteins. For this, a set of 103 stable isotope-labeled (SIS) and the corresponding natural (NAT) proteotypic peptides were used as internal standards, which were synthesized and validated by the Omics lab at Skoltech, as reported previously [13]. The method was adapted from the commercial BAK-270 kit (MRM Proteomics Inc., Victoria, BC, Canada) [11,14]. Following the addition of the SIS peptides, solid-phase extraction was used to clean up the samples, which were then reconstituted in 34 µL of 0.1% formic acid for LC-MS analysis.

To ensure analytical quality, an eight-point calibration curve (levels A–H) and three levels of quality control (QC-A, -B, -C) samples were prepared in accordance with established guidelines [14,45]. These were created by spiking a bovine serum albumin digest surrogate matrix with defined amounts of SIS and level-specific NAT peptides. A pooled control plasma QC sample was also analyzed to correct for inter-batch variation. All calibration and QC samples underwent the same sample preparation process as the clinical samples.

### 4.3. Targeted Proteomic Analysis by MRM MS

A targeted LC-MS/MS approach based on multiple reaction monitoring (MRM) was employed for the quantitative analysis of 103 plasma proteins. All samples were analyzed in duplicate using an ExionLC™ UHPLC system (AB Sciex, Framingham, MA, USA) coupled with a SCIEX QTRAP 6500+ mass spectrometer (SCIEX, Redwood City, CA, USA). LC separation occurred on a Zorbax Eclipse Plus RRHD (Agilent Technologies, Santa Clara, CA, USA) C18 column (2.1 × 150 mm, 1.8 µm) with a 30 min linear gradient (solvent B from 2% to 45%) at a flow rate of 0.4 mL/min. All other basic parameters for the LC-MS method were adapted from previously published research [13,14,44]. A full list of MRM transitions (precursor/product ion pairs) is provided in Appendix A.

Quantitative processing of the MRM data was performed with Skyline software (v20.2.0.343; University of Washington, Washington, DC, USA). Peptide concentrations in the plasma samples (reported in fmol/μL) were determined from calibration curves constructed using a 1/(x·x) weighting factor for linear regression. To ensure data quality, all protein and peptide measurements were validated according to ICH Bioanalytical Method Validation guidelines [45].

### 4.4. Statistical Analysis

Statistical analysis was performed with the Python3 programming language with numpy, scipy, pandas, statsmodels, scikit-learn, and lifelines packages. To compare qualitative parameters between independent groups of individuals, the Fisher exact test (two-tailed) was used with the significance level α for *p*-values set to 0.05. The two-sided Mann–Whitney U test was used for comparing distributions of quantitative parameters between two independent groups, while the Kruskal–Wallis test was used for comparing distributions of quantitative parameters between more than two independent groups. To control for type I error, false discovery rate q-values were calculated using the Benjamini–Hochberg (BH) procedure, and we set a threshold of 0.05 to keep the positive false discovery rate below 5%.

The Kaplan–Meier estimator was used to analyze overall survival; groups were compared with the log-rank test. Cox proportional hazards models (penalizer = 0.1) were used to quantify associations with mortality (hazard ratios, HR) for clinical stage and for protein concentration strata.

For single-protein Healthy vs. Stage I summaries, ROC AUC was computed on raw abundances; the F1-maximizing threshold from the precision–recall curve was used to report precision, recall, and F1 (apparent performance).

Association between six preselected plasma proteins and binary clinical status was assessed using a binomial generalized linear model with logit link, fitted under an L2 (ridge) penalty. Protein abundances were standardized to zero mean and unit variance, and an intercept term was included. We applied nested stratified cross-validation (outer = 5 folds for unbiased testing; inner = 5 folds for tuning λ and estimating operating thresholds). Across an inner grid from 0.05 to 1.50, inner-CV sensitivity analyses showed a broad performance plateau for λ ≥ 0.3; consequently, λ = 0.9 was selected as an a priori default value.

On the held-out outer test folds, the ridge-penalized regression achieved AUC = 0.933 ± 0.063 (mean ± SD), essentially invariant to the λ-selection strategy due to the plateau. For an operating point, we also examined the Youden index (J = sensitivity + specificity − 1). When the Youden threshold was tuned on each outer fold, performance was Accuracy = 0.90 ± 0.09, Precision = 0.82 ± 0.17, Recall = 1.00 ± 0.00, and Specificity = 0.85 ± 0.14, with thresholds ranging 0.23–1.00 across folds. As expected, apparent (full-set) metrics were higher than outer-test estimates: at λ = 0.90, AUC = 0.965, Accuracy = 0.94, Precision = 0.92, Recall = 0.92, and Specificity = 0.95 at Youden threshold = 0.49, consistent with optimism when evaluating on the training set. Regression coefficients were exponentiated to odds ratios; coefficient uncertainty was quantified via nonparametric bootstrapping (B = 300) with standardization inside each resample, yielding percentile 95% confidence intervals for odds ratios.

## 5. Conclusions

In this study, we have shown that alpha-1-acid glycoprotein 1, beta-2-microglobulin, plasma protease C1 inhibitor, leucine-rich alpha-2-glycoprotein, fibronectin, and cholinesterase may serve as potential biomarkers for the early diagnosis of PDAC. We propose a model that considers the levels of these proteins in blood plasma to distinguish between stage I PDAC patients and healthy donors with high accuracy and sensitivity. However, further research is needed to implement this model in clinical practice as a diagnostic tool for PDAC screening.

## Figures and Tables

**Figure 1 ijms-26-09219-f001:**
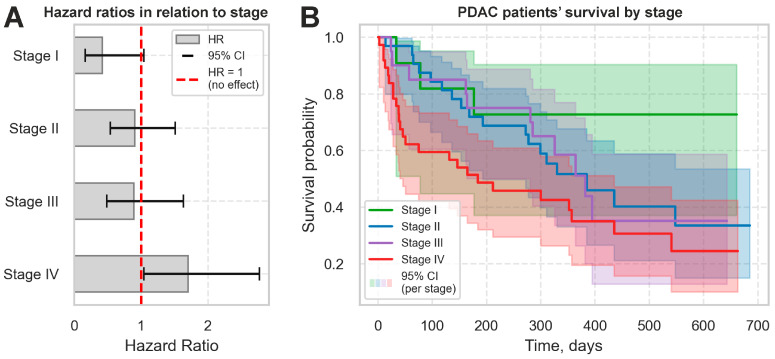
Association between PDAC stage and patients’ survival. (**A**) Cox model hazard ratios (HR [95% CI]) for clinical stages I–IV. (**B**) Kaplan–Meier curves of overall survival by stage; shaded bands reflect 95% CI. The only significant difference in survival rates has been found between stages I and IV (*p* = 0.022).

**Figure 2 ijms-26-09219-f002:**
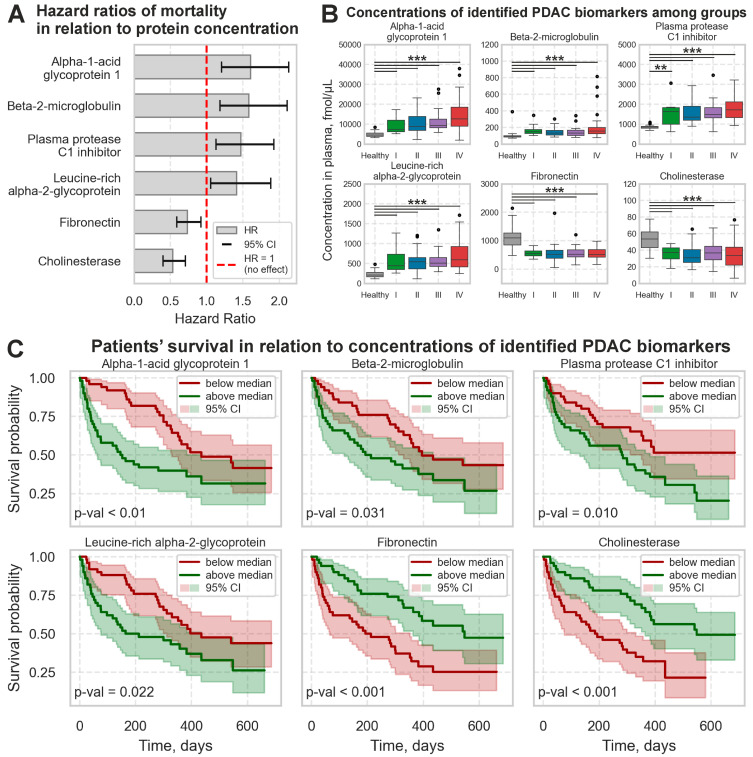
Six plasma proteins are significantly associated with the survival rates of PDAC patients. (**A**) Cox model HRs per 1 SD increase in protein concentration with 95% CI. (**B**) Distributions of plasma concentrations in healthy controls and PDAC patients at stages I–IV. (**C**) Kaplan–Meier curves for PDAC patients dichotomized at the median concentration of each analyzed protein. **—*p* < 0.01, ***—*p* < 0.001.

**Figure 3 ijms-26-09219-f003:**
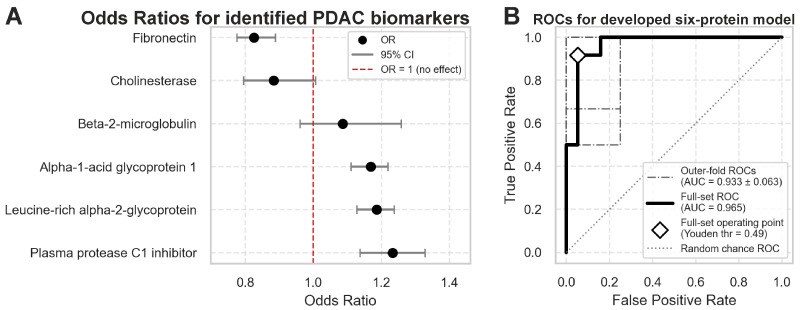
Regression model using six proteins to discriminate healthy controls vs. stage I PDAC. The model has been generated as described in Materials and Methods. (**A**) Odds ratios per 1 SD increase in each analyzed protein with 95% bootstrap CI. (**B**) ROC curves for the generated models on cross-validation folds and on the full dataset.

**Table 1 ijms-26-09219-t001:** Clinical parameters of the experimental groups.

Parameter	Healthy Controls (n = 19)	PDAC Patients				*p* *
		Stage I (n = 11)	Stage II (n = 35)	Stage III (n = 24)	Stage IV (n = 42)	
**Sex, % of males**	42.1	45.5	48.6	41.7	50.0	n.a.
**Age, years**	59 [57.5–67.5]	67.9 [65.6–74.8]	67.3 [59.7–70.8]	63.4 [57.7–71.8]	66.4 [61.9–72.0]	0.4290
**BMI, kg/m^2^**	28.4 [26.4–31.5]	26 [24.2–30.7]	25.2 [22.9–27.7]	23.7 [22.2–28.9]	25.6 [22.6–29.6]	0.0723
**WBC, 10^9^/L**	6.5 [6.4–8.1]	5.7 [4.7–6.9]	7.1 [5.8–9.0]	7.4 [5.8–9.0]	8 [6.4–10.3]	0.2084
**RBC, 10^12^/L**	4.7 [4.6–5.0]	4.5 [4.3–4.8]	4.2 [3.9–4.5]	4.3 [4.0–4.7]	4.2 [3.9–4.4]	0.0020
**Platelets, 10^9^/L**	206 [187.5–249.0]	228 [205.0–285.5]	228 [178.5–372.5]	216.5 [173.0–327.3]	246 [201.3–298.0]	0.8768
**Hemoglobin, g/L**	140 [136.0–146.0]	134 [123.0–139.5]	129 [118.0–134.5]	126 [109.5–136.0]	127.5 [111.8–135.0]	0.0025
**Total protein, g/L**	71 [69.0–74.0]	69.6 [66.3–73.9]	67.2 [61.6–71.5]	71.3 [68.5–73.4]	68.9 [64.5–73.3]	0.1030
**Creatinine, µmol/L**	84 [68.5–100.5]	82.3 [74.5–101.4]	73.1 [60.5–84.3]	78 [70.3–93.0]	85.8 [72.9–101.6]	0.0489
**Total bilirubin, µmol/L**	10.1 [7.9–12.4]	19.7 [14.4–38.0]	19.8 [11.2–29.9]	16.5 [13.9–28.6]	12.1 [10.2–20.5]	0.0020
**ALT, U/L**	22 [19.0–30.0]	38 [20.5–50.5]	31.7 [15.2–58.4]	28.1 [21.6–43.9]	21.9 [16.9–49.4]	0.5790
**AST, U/L**	31 [21.0–32.5]	29.3 [26.1–41.7]	28.2 [21.2–54.9]	24.4 [20.2–44.2]	31.3 [20.4–51.0]	0.7065
**Known outcomes, n**	19	11	33	20	37	n.a
**Lethal outcomes, n (%)**	0 (0.0)	3 (27.3)	19 (57.6)	11 (55.0)	25 (67.6)	n.a

* *p*-values calculated using Kruskal–Wallis test with Benjamini–Hochberg FDR correction. Values are presented as median [interquartile range] if not otherwise specified. BMI—body mass index, WBC—white blood cells, RBC—red blood cells, ALT—alanine aminotransferase, AST—aspartate aminotransferase, and n.a.—not applicable.

**Table 2 ijms-26-09219-t002:** Potential early biomarkers of PDAC among plasma proteins.

Protein	Concentration in Plasma, fmol/µL ^#^		Threshold Value, fmol/µL ^#^	Adjusted *p*-Value *	ROC AUC	Precision	Recall
	Healthy Controls	Stage I PDAC Patients					
Insulin-like growth factor-binding protein 3	153.95 [148.4–174.0]	76.895 [60.8–112.3]	125.02	0.00468	0.923	0.900	0.818
Beta-2-microglobulin	93.922 [82.2–99.8]	148.83 [127.3–175.2]	125.29	0.00468	0.919	0.900	0.818
Complement C1q subcomponent subunit C	217.6 [198.0–225.3]	346.35 [276.6–382.2]	263.96	0.00468	0.914	0.818	0.818
Fibronectin	1096.7 [851.9–1275.2]	544.09 [470.8–638.4]	828.81	0.00468	0.904	0.733	1.000
Complement component C9	350.28 [321.9–445.9]	613.81 [489.2–715.0]	458.12	0.00468	0.895	0.688	1.000
Sex hormone-binding globulin	25.669 [17.5–41.0]	75.003 [43.0–82.5]	35.551	0.00468	0.895	0.688	1.000
Plasma protease C1 inhibitor	849.39 [790.7–881.0]	1636.4 [1003.3–1829.6]	1144.5	0.00468	0.895	0.714	0.909
Leucine-rich alpha-2-glycoprotein	207.79 [164.5–275.7]	442.98 [347.2–730.9]	345.63	0.00468	0.895	0.714	0.909
Lysozyme C	45.724 [39.9–58.9]	70.473 [62.0–104.0]	55.461	0.00495	0.890	0.688	1.000
Carboxypeptidase N catalytic chain	83.899 [75.6–95.8]	127.83 [102.3–164.1]	98.346	0.00528	0.885	0.647	1.000
Alpha-1-acid glycoprotein 1	4505.4 [3865.7–5494.1]	7303.8 [6086.6–11,986.5]	5119	0.00715	0.871	0.647	1.000
Alpha-1-antitrypsin	18,071 [16,772.0–21,351.5]	23,871 [22,170.5–33,500.5]	20310	0.00715	0.871	0.647	1.000
Cholinesterase	53.523 [43.2–62.0]	37.244 [29.5–42.7]	47.82	0.00774	0.866	0.800	0.727
Vasorin	8.2042 [6.5–9.0]	11.324 [9.2–16.1]	11.279	0.00976	0.856	1.000	0.636
Fibrinogen alpha chain	15,715 [14,400.5–18,832.0]	21,671 [19,314.0–27,096.0]	19839	0.01227	0.847	0.875	0.636
Fibrinogen gamma chain	11,700 [10,938.5–13,812.5]	14,727 [13,575.0–19,831.5]	12833	0.01586	0.833	0.889	0.727
Thyroxine-binding globulin	162.4 [146.4–181.0]	220.23 [182.2–252.9]	217.62	0.01586	0.833	0.889	0.727
Complement C5	382.31 [338.1–461.0]	546.9 [448.2–595.8]	503.44	0.01649	0.828	0.588	0.909
Complement C1r subcomponent-like protein	53.314 [49.8–58.9]	66.84 [57.5–75.7]	54.662	0.01979	0.818	0.625	0.909
Complement factor B	1523.9 [1344.5–2004.6]	2602.5 [1849.4–2997.6]	2602.5	0.01979	0.818	0.625	0.909

* *p*-values calculated using Mann–Whitney U test with Benjamini–Hochberg FDR correction, ^#^ Values are presented as median [interquartile range], ROC AUC—area under the ROC curve.

## Data Availability

The data presented in this study are available in the article and Appendix A.

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
