# Peer review of "Multiplexed Quantitation of Plasma Proteins by Targeted Mass Spectrometry for Early Diagnosis of Pancreatic Ductal Adenocarcinoma"

_ijms, 2025, doi:10.3390/ijms26189219_

Round 1

Reviewer 1 Report

Comments and Suggestions for Authors

The manuscript entitled “Multiplexed quantitation of plasma proteins by targeted mass spectrometry for early diagnosis of pancreatic ductal adenocarcinoma” describes the use of targeted mass spectrometry to quantify over one hundred plasma proteins in patients with PDAC and healthy controls. From this analysis, the authors identify six proteins associated both with disease presence and with survival, and they construct a predictive model capable of distinguishing early-stage PDAC from controls with high accuracy. The study is clearly written and, importantly, goes beyond a purely descriptive approach by linking protein levels to clinically relevant outcomes. Nevertheless, I have a few comments that could help to further strengthen its clarity and impact.

  1. The abstract would benefit from a clearer structure. Briefly outlining the background, specifying the design (sample size, groups), highlighting the key results with appropriate statistics, and ending with a balanced conclusion that also notes limitations.
  2. The manuscript lacks “Results” subsections, which makes it difficult for readers to follow the progression of findings. Adding descriptive headings would improve readability.
  3. More detail on patient selection is needed, including exclusion criteria and how potential confounders were handled.
  4. Figures lack sufficient legend detail, which makes them hard to interpret independently of the text.
  5. The discussion of clinical applicability could be deepened. The manuscript notes that combinations such as CA19-9 with other proteins have shown strong performance. However, it would be helpful to directly compare the six-protein panel to these reported combinations and comment on whether it may offer incremental value or complementary utility.

Author Response

First of all, together with co-authors we would like to thank the Reviewer for the time devoted to thorough examination of our work and valuable comments. Point-to-point response to the Reviewer’s comments is presented below.

Reviewer 1

The manuscript entitled “Multiplexed quantitation of plasma proteins by targeted mass spectrometry for early diagnosis of pancreatic ductal adenocarcinoma” describes the use of targeted mass spectrometry to quantify over one hundred plasma proteins in patients with PDAC and healthy controls. From this analysis, the authors identify six proteins associated both with disease presence and with survival, and they construct a predictive model capable of distinguishing early-stage PDAC from controls with high accuracy. The study is clearly written and, importantly, goes beyond a purely descriptive approach by linking protein levels to clinically relevant outcomes. Nevertheless, I have a few comments that could help to further strengthen its clarity and impact.

Point 1: The abstract would benefit from a clearer structure. Briefly outlining the background, specifying the design (sample size, groups), highlighting the key results with appropriate statistics, and ending with a balanced conclusion that also notes limitations.

Response 1: The abstract has been edited in accordance with the Reviewer’s comments.

Point 2: The manuscript lacks “Results” subsections, which makes it difficult for readers to follow the progression of findings. Adding descriptive headings would improve readability.

Response 2: According to the Reviewer’s comment, we added subheadings to the Results section.

Point 3: More detail on patient selection is needed, including exclusion criteria and how potential confounders were handled.

Response 3: In the amended manuscript we added the exclusion criteria and discussed the confounders managing issue (page 2, lines 70-72; pages 4-5, lines 269-272).

Point 4: Figures lack sufficient legend detail, which makes them hard to interpret independently of the text.

Response 4: The figures have been edited according to the Reviewer’s comments.

Point 5: The discussion of clinical applicability could be deepened. The manuscript notes that combinations such as CA19-9 with other proteins have shown strong performance. However, it would be helpful to directly compare the six-protein panel to these reported combinations and comment on whether it may offer incremental value or complementary utility.

Response 5: According to the issues raised by the Reviewer, we expanded the discussion and added into the revised manuscript text comparison of the developed six-protein panel with the reported approaches (page 3, lines 182-191, 212-213; page 4, lines 242-266).

Reviewer 2 Report

Comments and Suggestions for Authors

This is a well-designed and timely study that addresses a critical unmet need in pancreatic ductal adenocarcinoma (PDAC) diagnostics. The use of targeted mass spectrometry (MRM) to identify and validate a panel of six plasma proteins for early-stage PDAC detection is methodologically sound and clinically relevant. The manuscript is clearly written, and the statistical analyses are rigorous. The proposed biomarker panel shows excellent discriminatory power (AUC = 0.965) between stage I PDAC and healthy controls. However, several issues should be addressed before publication.

Major Comments

    • The healthy control group (n=19) is significantly smaller than the PDAC group (n=113). This imbalance may affect the generalizability and statistical power of the results. The authors should justify the sample size and consider discussing this as a limitation.
    • It would be beneficial to include a validation cohort to confirm the robustness of the six-protein panel.
    • The study lacks an independent validation cohort. The performance of the model should be tested on an external dataset to ensure its reliability and clinical applicability.
    • Several of the identified proteins (e.g., beta-2-microglobulin, leucine-rich alpha-2-glycoprotein) are known to be elevated in other cancers and inflammatory conditions. The authors should address the potential lack of specificity and discuss how the combination of markers may mitigate this issue.
    • Including comparisons with other cancer types or benign pancreatic diseases (e.g., chronic pancreatitis) would strengthen the specificity claims.
    • While the model shows high accuracy, its clinical utility remains to be demonstrated. The authors should discuss how this panel would be integrated into current diagnostic pathways (e.g., alongside imaging or CA19-9).
    • Cost, scalability, and turnaround time of the MRM assay should be briefly discussed.
    • The use of a ridge regression model is appropriate, but the lack of cross-validation or external validation is a concern. The authors should clarify whether the reported performance metrics are based on the training set only.
    • The choice of penalizer (λ = 0.9) should be justified, and sensitivity analysis around this parameter would be beneficial.

Minor Comments

    • The interquartile ranges in Table 2 appear to contain typographical errors (e.g., “89.9-58.9” for Lysozyme C). Please verify all values.
    • Figures 1–3 are informative but could be improved with better labeling and resolution. Error bars and confidence intervals should be clearly indicated.
    • The discussion would benefit from a more direct comparison with existing multi-marker panels (e.g., the 29-protein signature) and a clearer justification for why a 6-protein panel is superior or more practical.
    • Supplementary Table S1 and S2 are referenced but not provided in the submission. These should be made available for review.
    • Please ensure all abbreviations are defined at first use (e.g., SIS, NAT, CPL)

Author Response

First of all, together with co-authors we would like to thank the Reviewer for the time devoted to thorough examination of our work and valuable comments. Point-to-point response to the Reviewer’s comments is presented below.

Reviewer 2

This is a well-designed and timely study that addresses a critical unmet need in pancreatic ductal adenocarcinoma (PDAC) diagnostics. The use of targeted mass spectrometry (MRM) to identify and validate a panel of six plasma proteins for early-stage PDAC detection is methodologically sound and clinically relevant. The manuscript is clearly written, and the statistical analyses are rigorous. The proposed biomarker panel shows excellent discriminatory power (AUC = 0.965) between stage I PDAC and healthy controls. However, several issues should be addressed before publication.

Major Comments

Point 1: The healthy control group (n=19) is significantly smaller than the PDAC group (n=113). This imbalance may affect the generalizability and statistical power of the results. The authors should justify the sample size and consider discussing this as a limitation.

Response 1: Indeed, in the current study we enrolled 113 patients with PDAC and 19 healthy controls. However, all comparisons were performed between healthy controls and PDAC patients at different stages. In this respect, as it was shown in Table 1, the group sizes were comparable: healthy controls (n = 19), PDAC patients at stage I (n = 11), stage II (n = 35), stage III (n = 24), and stage IV (n = 42). But we agree with the Reviewer, that the groups still had relatively small number of participants. Nevertheless, within the study we controlled the false discovery rate, as well as the significance levels were found to be rather high (adjusted p-values ~10-3). All this suggests that the results obtained are pronounced enough to be detected even on a relatively small number of samples, and therefore they are likely to remain reliable when the size of the analyzed cohort is expanded. We added the discussion of this issue in the manuscript text and pointed as a limitation (page 4, lines 250-258).

Point 2: It would be beneficial to include a validation cohort to confirm the robustness of the six-protein panel.

Point 3: The study lacks an independent validation cohort. The performance of the model should be tested on an external dataset to ensure its reliability and clinical applicability.

Point 4: Several of the identified proteins (e.g., beta-2-microglobulin, leucine-rich alpha-2-glycoprotein) are known to be elevated in other cancers and inflammatory conditions. The authors should address the potential lack of specificity and discuss how the combination of markers may mitigate this issue.

Point 5: Including comparisons with other cancer types or benign pancreatic diseases (e.g., chronic pancreatitis) would strengthen the specificity claims.

Point 6: While the model shows high accuracy, its clinical utility remains to be demonstrated. The authors should discuss how this panel would be integrated into current diagnostic pathways (e.g., alongside imaging or CA19-9).

Response 2-6: We thank the Reviewer for the valuable comments. Indeed, the developed panel needs subsequent validation and analysis for specificity as we indicated in the Discussion section as limitations of the study. Unfortunately, enrolling a large number of participants and testing several hundred proteins using targeted mass spectrometry presents a significant challenge in terms of both financial and labor costs. Therefore, current study aimed at identification among many plasma proteins a limited number of potential PDAC biomarkers, which can be further studied more comprehensively. In the current study we identified six proteins as the most promising biomarkers for PDAC diagnosis and prognosis, and next we are going to study them for specificity/sensitivity using expanded cohort of PDAC patients and patients with a wide range of pathologies, other than PDAC. Additionally, we plan to analyze the performance of the individual six protein panel and in combination with other biomarkers, particularly CA19-9, for early diagnosis of PDAC. Performing a study with hundreds of proteins is a challenging task, but it is more feasible with the identified six proteins. Therefore, the results presented in the current manuscript support the idea of conducting such a study. We discussed this issue in the amended text of the manuscript (page 4, lines 250-266).

Point 7: Cost, scalability, and turnaround time of the MRM assay should be briefly discussed.

Response 7: We would like to thank the Reviewer for this important issue. Indeed, MRM is rather cost-effective and highly scalable for targeted analysis. A single method can robustly quantify dozens to hundreds of analytes simultaneously in one injection. This high multiplexing capability, combined with high-throughput liquid chromatography (e.g., short gradients, 5-10 min) and automated sample processing, allows a single instrument to analyze hundreds of samples per week. This makes it a very powerful tool for the cohort studies (e.g., clinical validation or biomarker verification). We added the discussion of this issue according to the Reviewer’s comment (page 3, lines 193-203).

Point 8: The use of a ridge regression model is appropriate, but the lack of cross-validation or external validation is a concern. The authors should clarify whether the reported performance metrics are based on the training set only.

Point 9: The choice of penalizer (λ = 0.9) should be justified, and sensitivity analysis around this parameter would be beneficial.

Response 8-9: We agree that single-fit, training-set metrics tend to be over-optimistic. To address it, we replaced training-set estimates with nested stratified cross-validation. Results were stable across a broad range of penalties, supporting our pre-specified λ as the default. We switched to reporting averaged cross-validated metrics, and apparent full-set training metrics are now shown in context of these estimates. We would like to emphasize that this remains a proof-of-concept study, and confirmation on larger, independent cohorts will be essential part of further research. We changed the manuscript text in accordance with the performed analysis (page 2, lines 135-138; page 6, lines 338-356).

Minor Comments

Point 10: The interquartile ranges in Table 2 appear to contain typographical errors (e.g., “89.9-58.9” for Lysozyme C). Please verify all values.

Response 10: According to the Reviewer’s comment we have double checked the values presented in Table 2.

Point 11: Figures 1–3 are informative but could be improved with better labeling and resolution. Error bars and confidence intervals should be clearly indicated.

Response 11: The figures have been edited according to the Reviewer’s comments.

Point 12: The discussion would benefit from a more direct comparison with existing multi-marker panels (e.g., the 29-protein signature) and a clearer justification for why a 6-protein panel is superior or more practical.

Response 12: In the revised manuscript we have discussed the issue concerning the comparison of our six-protein panel with the previously reported biomarker-based approaches for early-stage PDAC diagnosis (page 3, lines 182-191; page 3, lines 212-213; page 4, lines 242-249).

Point 13: Supplementary Table S1 and S2 are referenced but not provided in the submission. These should be made available for review.

Response 13: Supplementary Tables S1 and S2 were uploaded during the initial submission, but for some unknown reason they seemed to have been missed. We apologize for this unfortunate misunderstanding and have uploaded the tables once more.

Point 14: Please ensure all abbreviations are defined at first use (e.g., SIS, NAT, CPL)

Response 14: According to the Reviewer’s comment, we have checked that the abbreviations are defined at the first use.

Round 2

Reviewer 2 Report

Comments and Suggestions for Authors

The authors have addressed my previous comments; therefore, the manuscript can be considered for publication